# Brief Communication: Update on the GPS Reflection Technique for Measuring Snow Accumulation in Greenland

Kristine M. Larson[1], Michael MacFerrin[2], Thomas Nylen[3]

[1]Department of Aerospace Engineering Sciences, University of Colorado, Boulder, CO, 80309, USA
[2]CIRES, University of Colorado, Boulder, CO 80309, USA
[3]UNAVCO, 6350 Nautilus Drive, Boulder, CO 80301, USA

*Correspondence to*: Kristine M. Larson (kristinem.larson@gmail.com)

**Abstract.** GPS Interferometric Reflectometry (GPS-IR) is a technique that can be used to measure snow accumulation on ice sheets. The footprint of the method (~1000 m^2) is larger than many other in situ methods. A long-term comparison with hand-measurements yielded an accuracy assessment of 2 cm. Depending on the placement of the GPS antenna, these data are also sensitive to firn density. The purpose of this short note is to make public GPS-IR measurements of snow accumulation for four sites in Greenland, compare these records with in situ sensors, and to make available open source GPS-IR software to the cryosphere community.

## 1 Introduction

GPS Interferometric Reflectometry (GPS-IR) was first described and validated for measuring seasonal snow accumulation in the western U.S. (Larson et al., 2009; McCreight et al., 2014). GPS-IR takes advantage of the fact that reflected GPS signals at low elevation angles from natural surfaces such as snow and ice are minimally rejected by geodetic antennas (Figure 1). The interference between the direct and reflected GPS signals produces a characteristic pattern in Signal to Noise Ratio (SNR) data that can be used to retrieve the height of the GPS antenna phase center above the top of the snow/ice surface. These vertical reflection distances (also called reflector heights, or RH) are estimated for every rising and setting GPS satellite arc; a daily average RH is then computed. The daily RH measurement has a footprint of ~ 1000 m^2 at sites where the antenna height is ~2 meters.

Three GPS receivers were installed on the interior of the Greenland ice sheet in 2011 by the GLISN project (GreenLand Ice Sheet monitoring Network, Clinton et al., 2005, Figure 2). The original scientific application of these data was to precisely measure the three-dimensional movement of the ice sheets. Larson et al. (2015, hereafter L2015) showed that GPS-IR could be combined with GPS-derived vertical coordinates to provide information about both snow accumulation and firn density.

L2015 summarized the GPS-IR technique and presented analysis of GPS-IR results for the period 2011-2014. Comparisons with another instrument (ultrasonic snow depth sensor) and regional atmospheric climate models were limited and qualitative. Since that time the GPS-IR technique has been successfully used in Antarctica (Siegfried et al., 2017; Shean et al., 2017). The former also compared GPS-IR retrievals with manual snow height measurements, yielding an accuracy assessment of 2 cm. Since the GLISN deployment began, a new GPS-IR site, SMM3, has been added in Greenland at Summit Camp.

The purpose of this brief communication is:

1. Present and archive GPS-IR results for these four sites in Greenland.
2. Compare GPS-IR snow accumulation records with other in situ datasets.
3. Provide short descriptions and links to publicly available GPS-IR software for the cryosphere community to use.

## 2 GPS Data

The original GLISN sites in Greenland are located at the Dye 2, Ice South Station, and NEEM field camps (GLS1, GLS2, GLS3). GLS3 was moved to a new monument in 2012. A fourth GPS reflection site was installed at Summit Camp in summer of 2017 (SMM3). Each GPS receiver is a dual-frequency carrier phase geodetic-quality unit. At the GLISN sites, the antenna is mounted to a pole which is attached to a plywood base and then buried 0.5-1.5 meters below the surface. At installation the pole was ~3 meters above the ice surface. At SMM3, the antenna is attached at the top of a 16.5m Rohn tower, which when installed had approximately 0.5m of the tower below the surface. The GPS data for the GLISN sites are telemetered on an hourly or daily basis via Iridium modems to the UNAVCO archiving facility. Raw GPS data from all four sites are archived at UNAVCO and freely available to the public. For this study, 15 second GPS sampling rates and the L1 SNR GPS data were used.

## 3 Summary of Archived Results

Here we have archived the RH measurements with daily position results computed by the Nevada Geodetic Laboratory (2019). Figure 1 describes the similarities and differences between the two kinds of GPS measurements. RH measures the distance between the GPS antenna phase center and the top of the ice/snow surface. The geocentric vertical coordinates measure how the pole moves with respect to the center of the Earth. Both measurements are sensitive to the length of the pole that connects the antenna to the base. When the pole is extended, those pole extensions (which are identical for the two kinds of measurements) must be corrected in both data sets.

At GLS1 only the RH are shown (Figure 3A). The RH measurements clearly show when the pole was lengthened, in 2016 and 2017. Elevation of the snow surface (Figure 3B) is the mirror of the RH after the pole offsets are removed. At this site

the geocentric vertical coordinates are not used except to calculate the pole extensions. At GLS2 and GLS3 both RH and geocentric vertical coordinates are shown (Figures 3C-F). Both RH and geocentric verticals are sensitive to the length of the pole, so when the poles are extended there is an immediate and equal response in both measurements. Additionally, L2015 explained that the geocentric vertical changes reflect a combination of two effects: firn compaction between the surface and the antenna anchor point and vertical movement due to the local ice slope (derived from the GIMP elevation dataset, Howat et al, 2014). The second of these two effects is found to be small but non-negligible, 1.9 and 1.1 cm/yr downwards for GLS2 and GLS3, respectively. The RH are affected by both new snow fall/surface melt and firn compaction. At sites with significant snow compaction effects, L2015 suggested that the effects could be removed by subtracting the geocentric vertical positions from the RH. The latter would need to be adjusted by the local slope vertical velocity, which does not affect RH.  Figures 3D and F show this combination, RH – (geocentric verticals + local slope velocity), with a sign change so as to indicate surface elevation changes.

Figure 3G summarizes the results for Summit Camp. Unlike the GLISN sites, which are between 1 and 3 meters above the snow/ice surface, this GPS antenna was mounted on a 16.5-meter tall tower. The first several months of the data records show non-linear motion for both vertical positions and RH, which you would expect for a taller tower with a shallow installation and relatively dry firn. We used the data from November 2017 to November 2019 to detrend the RH in order to emphasize the accumulation and melt events. The archived result file for SMM3 includes both the raw RH and the geocentric vertical coordinates. Here we have chosen not to remove effects of snow compaction.

## 4 Comparison with In Situ Sensors

We compare the GPS-IR snow accumulation records at Dye 2 with two independent in situ snow accumulation sensors (Figure 3A). The first is a six-year record provided by the Greenland Climate Network, GC-Net (Steffen and Box, 2001). The GC-Net instrument is an ultrasonic snow depth sensor that has operated since 1995. It is ~2 km from the GPS site. It is estimated to be anchored at a depth of 15 m in 2011 (personal communication from Koni Steffen, 2014). The correlation between the two records is 0.993, and the standard deviation of the difference is 9.4 cm. The GPS-IR records between 2015.5 and 2019.5 are compared with another ultrasonic snow depth sensor ~500 meters from the GC-Net unit. It was installed as part of the NASA-funded Firn Compaction Verification and Reconnaissance (FirnCover) project. Similar to the GC-Net sensor, the correlation between the GPS-IR records and FirnCover is very strong (0.992) and the standard deviation of the difference is 9.9 cm. These comparisons demonstrate the fidelity of the GPS-IR records for long-term snow accumulation studies compared with established field measurements of snow depth. The 9.4 and 9.9 cm standard deviation of anomalies are less than or equal to the magnitude of wind-blown features such as sastrugi that can migrate underneath the relatively small footprints of the sonic-ranging sensors during their measurement periods.

## 5 GPS-IR Software

GPS-IR is based on extracting characteristic frequencies found in GPS SNR data. The general principles of GPS-IR and some sample datasets are provided by Roesler and Larson (2018). The code needed to apply these principles to new GPS datasets is now available at GitHub (Larson, 2019). These codes assume that the GPS data are available in the standard
format known as RINEX (Gurtner, 1994). Fortran code is available to read RINEX files and compute the needed GPS-IR parameters (i.e. satellite elevation and azimuth angles). GPS-IR analysis software to routinely compute reflector heights from these files is available on GitHub in both Matlab and Python, along with a user manual.

For cryosphere scientists who might have an interest in GPS-IR but don't have a routine need for it, a web app has been
developed that automatically computes reflector heights (https://gnss-reflections.org). The beta version of the app supports both data that have been archived at major GPS data centers and user-provided RINEX files (Larson, 2020). The GitHub site discussed in the previous paragraph provides a bash script that can be used to call the web app along with some examples.

## 6 Conclusions

GPS-IR is an accurate and precise method to measure snow accumulation on ice sheets. No modifications are needed to the
GPS equipment more typically used to measure accurate three-dimensional positions. Compared to ultra-sonic snow depth sensors, GPS-IR has a significantly larger footprint (~1000 m^2 for an antenna that is 2-meter above the snow/ice) and thus is more representative of a given region. GPS (and now GNSS) receivers are routinely operated in Greenland and Antarctica with excellent data retrieval records. We hope that the results shown here and the software described in this short note will lead to GPS-IR being more easily used by the cryosphere community.


**Code and data availability**

The code used in this study is freely available from GitHub, https://github.com/kristinemlarson.

**Supplement**

The supplement related to this article is available online at: (to be filled in by you).


**Author contributions**

KL analyzed the data, prepared the datafiles in the supplement, and made Figures 1 and 3. MM made Figure 2 and provided in situ datasets for comparisons. All authors contributed to the text of the paper.

**Competing interests**

The authors declare that they have no conflicts of interest.

**Acknowledgements**

We thank Konrad Steffen for providing the GC-Net in situ snow accumulation data for Dye 2. Additional acknowledgements are found in the supplementary information provided with the archived reflector height data. We thank Dean Childs and
Kevin Nikolaus at IRIS PASSCAL for answering questions and providing photographs. We thank the late John Wahr for inspiring this work. An editor and two reviewers provided helpful feedback on the manuscript. Radon Rosborough and Evan Crouch helped improve the GNSS reflectometry web app.

**Financial Support**


The FirnCover data are provided by awards NASA NNX10AR76G and NNX15AC62G, which also funded M. MacFerrin's work. K.M. Larson's work was supported by GeoForschungsZentrum Potsdam and a research fellowship from the Alexander von Humboldt Foundation. NSF OPP 1304011 supported the installation and maintenance of the GLISN GPS sites by IRIS and UNAVCO and archiving of the data by UNAVCO.

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

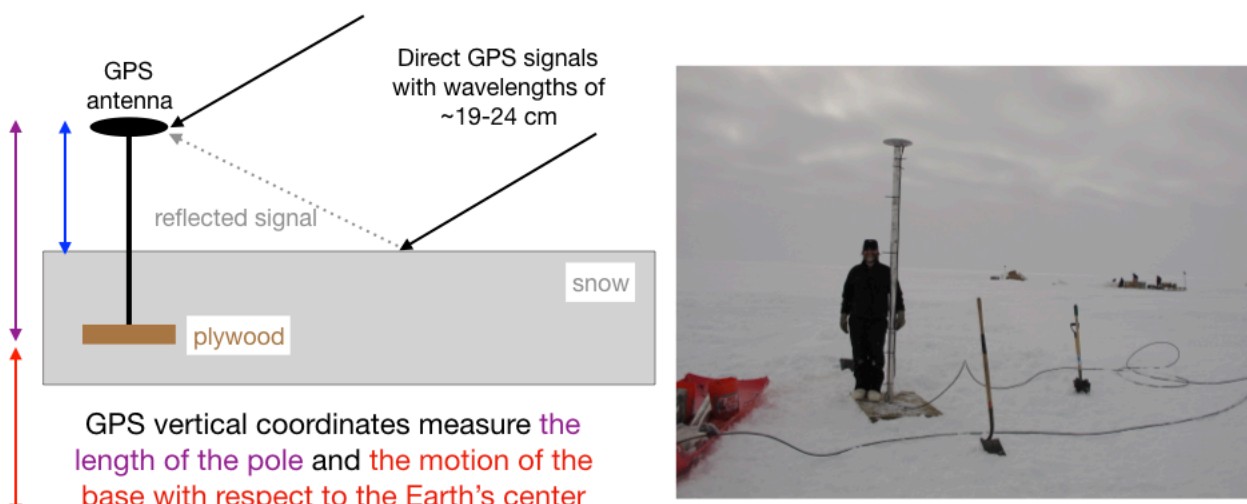

Figure 1. On the left GPS-IR geometry is shown along with a drawing of a typical GLISN installation. The reflected signal (in gray) interferes with the direct signal (black), which creates an interference pattern directly related to the reflector height (blue). The latter is defined as the distance between the GPS antenna phase center and the top of the ice/snow surface. The GPS vertical coordinates are defined relative to the center of the Earth. On the right is a photograph of GPS station GLS1 in 2011. The GPS antenna is attached to a monument made of plywood, buried 1.5 meters below the surface at installation.

Photo courtesy of IRIS.

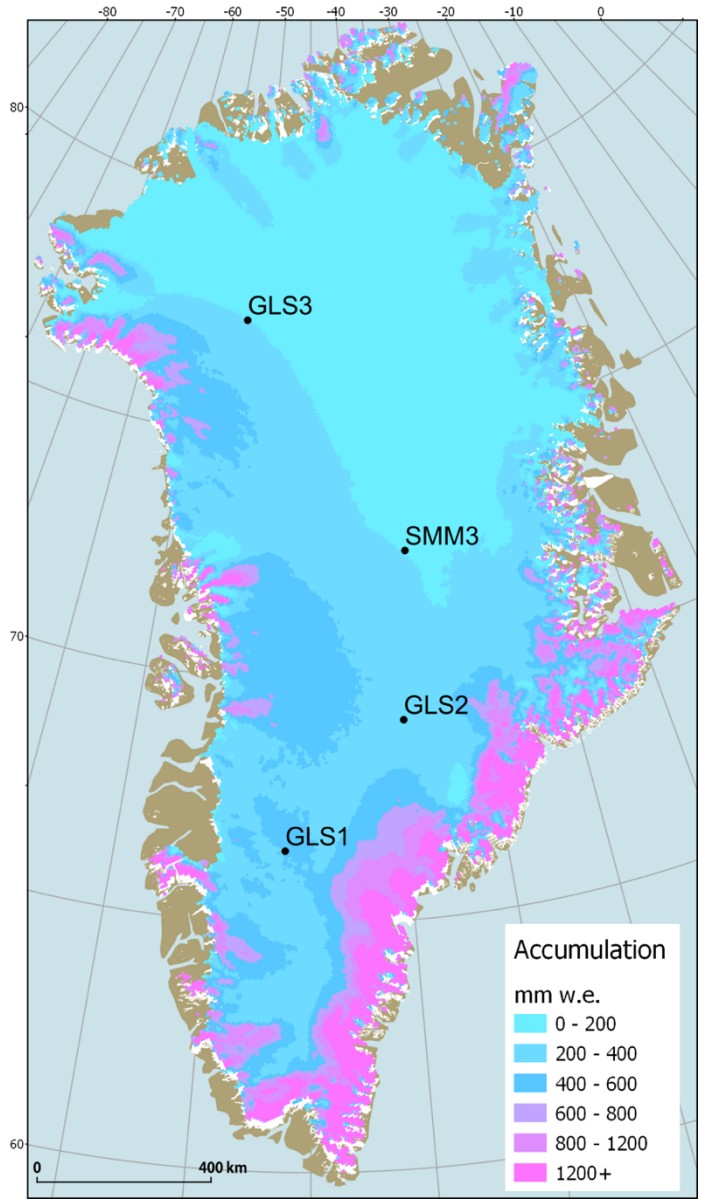


Figure 2. Map locations of the GLISN GPS sites and SMM3 (Summit Camp). Mean annual snow accumulation rates (water equivalent) are also shown (Mottram et al., 2017).


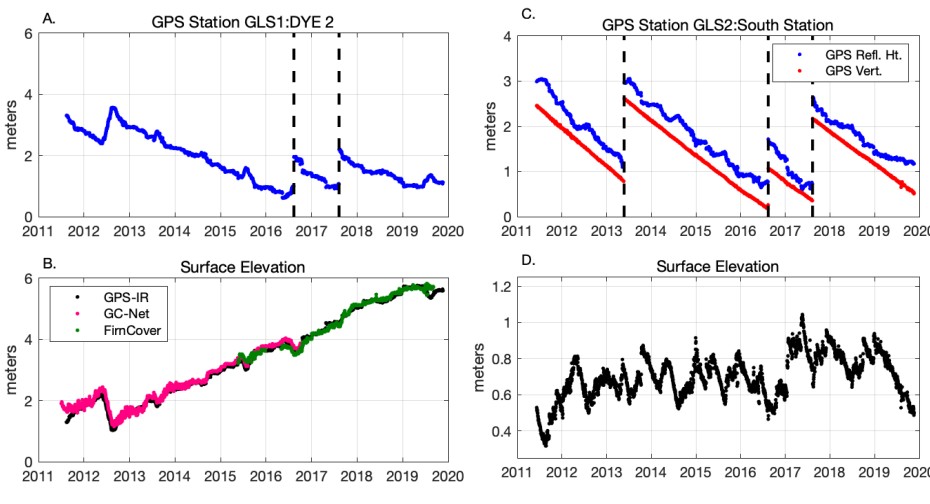

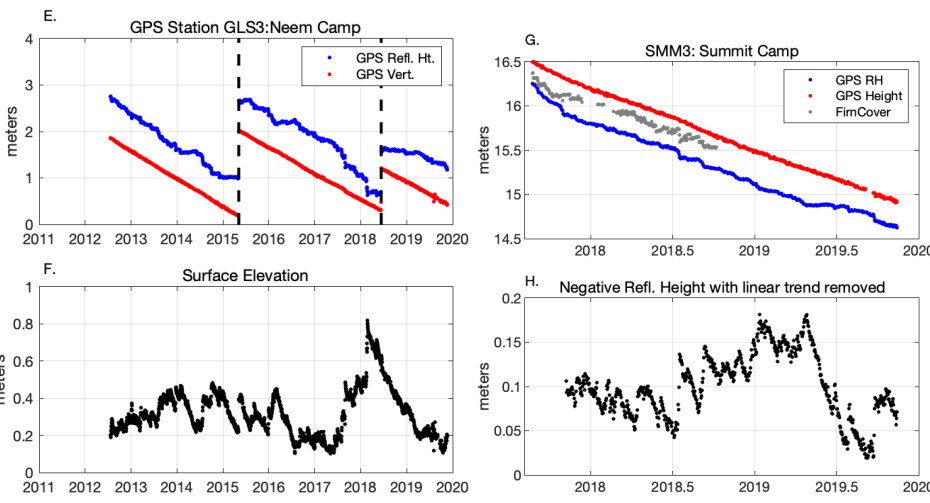

Figure 3A: GLS1 reflector heights; 3B. surface elevation for GLS1 compared to *in situ* measurements from GC-Net and FirnCover; 3C: GLS2 reflector heights (blue) and geocentric verticals (red); 3D: surface elevation for GLS2; 3E: GLS3 reflector heights (blue) and geocentric verticals (red); 3F: surface elevation for GLS3; 3G: SMM3 reflector heights (blue) and geocentric verticals (red), and in situ measurements from FirnCover; 3H: SMM3 reflector heights with linear trend removed and sign change. All GPS verticals are defined with respect to the center of the Earth, and thus a constant has been removed before plotting them.