# Peer review of "Brief Communication: Update on the GPS Reflection Technique for Measuring Snow Accumulation in Greenland"

_The Cryosphere, 2019_

## Referee Comment (RC1) · Anonymous Referee #1 · 25 Mar 2020

This manuscript by Larson et al. is almost fit for final publication in TC. However, for the sake of readability, I would recommend to reiterate the principle of GPS-IR in the introductory section. It is currently in section 3, but it would make sense to me to move this to the start of the paper. My suggestion would be to insert this as a first paragraph of section 1, or after the sentence in page 1 line 22.

page 3 line 66: can you explain how these vertical velocities due to the ice slope were derived?

page 3 line 74: can you clarify what you would expect? Does the combination of a shallow anchor and a tall tower cause additional settling of the tower? Please clarify

whether the non-linear effects are an artefact of the new installation.

page 3 line 89: equal-to –> equal to

page 3 line 90: relative-small –> relatively small

page 4 line 99: Suggest to add that a user manual is available at GitHub as well.
* * *

---

## Referee Comment (RC2) · Anonymous Referee #2 · 27 Apr 2020

This manuscript by Larson et al. is ready for final publication in TC if the Anonymous Referee #1's (RC1) minor suggestions are made. It is encouraging to see how GPS-IR is being used to measure snow accumulation on ice sheets.

---

## Author Comment (AC2) · 4 May 2020

Reviewer 2 asked us to comply with edits of Reviewer 1. We have done so and these are supplied in previous comment.

---

## Author Response (AR1)

**Brief Communication: Update on the GPS Reflection Technique for Measuring Snow Accumulation in Greenland**

Per the suggestion of the editor, I removed Table 1. The information from the table is available in the text.

Other small changes: I found that I had defined SNR twice, so removed the second time. I had also apparently stated that the GLISN sites were installed in 2011 twice, so also removed that repetitiveness.

These other responses were also uploaded to the web site, but for completeness, I have copied them here.
* * *
Thank you to the reviewers and the original editor for helping us to improve the manuscript. I have followed the instructions online for the format of this response, that it shall be structured in a clear and easy-to-follow sequence: (1) comments from Referees, (2) author's response, (3) author's changes in manuscript.

Reviewer 1.

*This manuscript by Larson et al. is almost fit for final publication in TC. However, for the sake of readability, I would recommend to reiterate the principle of GPS-IR in the introductory section. It is currently in section 3, but it would make sense to me to move this to the start of the paper. My suggestion would be to insert this as a first paragraph of section 1, or after the sentence in page 1 line 22.*

We agree with the reviewer's recommendation; we have chosen to make it the first paragraph of section 1. Please note that because of this change Figures 1 and 2 had to have their order changed.

*page 3 line 66: can you explain how these vertical velocities due to the ice slope were derived?*

We used the analysis from L2015, as stated from page 107:

Figure 6 shows the direction of each horizontal velocity vector superimposed on local surface elevation maps extracted from the GIMP Greenland elevation dataset (Howat and others, 2014). In each case, the velocity vector is directed downslope, as is expected for flowing ice. Columns 6

and 7 of Table 1 show that at each site the directions of the velocity and the downslope gradient agree to within a few degrees. The slopes of the GIMP topography (vertical drop per horizontal distance) averaged over 5 km of each GPS site are 0.0053, 0.0023 and 0.0019 at GLS1, GLS2 and GLS3 respectively. By combining these slopes with the horizontal velocity values, we obtain estimates of $V_0$, the downward vertical velocity of the surface caused by the downhill flow of the ice sheet, shown in row 10 of Table 1.

We have added the GIMP dataset to the paper with the Howat reference.

*page 3 line 74: can you clarify what you would expect? Does the combination of a shallow anchor and a tall tower cause additional settling of the tower? Please clarify whether the non-linear effects are an artefact of the new installation.*

The dry firn at Summit Camp settles far more easily under a load than previously-wetted-and-refrozen firn in other places (such as Dye-2, etc). A 16.5 m tower atop a shallow platform will cause the firn to compress underneath it for a length of time after installation. The need to tighten guy-wires as the towers settle relative to guy-wire anchors is even found for much shorter towers at Summit Camp (e.g. 4 meter towers). We thus agree with the reviewer that the non-linear and accelerated snow-depth effects on the 16.5 m installation are, in fact, be an artifact of the installation at least for a short while.   Clarification added.

*page 3 line 89: equal-to –> equal to*

Fixed

*page 3 line 90: relative-small –> relatively small*

Fixed

*page 4 line 99: Suggest to add that a user manual is available at GitHub as well.*

Added. I also fixed the name of the website, as the user is now directed to the main area, https://gnss-reflections.org rather than a sub-domain.

*Reviewer 2.*
*Agreed with reviewer 1 comments.*

Please note that the acknowledgements have been updated to thank both the editor and reviewers.